# CoMemo: LVLMs Need Image Context with Image Memory

**Shi Liu** [* 1]  **Weijie Su** [* † ✉ 1]  **Xizhou Zhu** [2 1]  **Wenhai Wang** [3 1]  **Jifeng Dai** [2 1]

## Abstract

Recent advancements in Large Vision-Language Models built upon Large Language Models have established aligning visual features with LLM representations as the dominant paradigm. However, inherited LLM architectural designs introduce suboptimal characteristics for multimodal processing. First, LVLMs exhibit a bimodal distribution in attention allocation, leading to the progressive neglect of middle visual content as context expands. Second, conventional positional encoding schemes fail to preserve vital 2D structural relationships when processing dynamic high-resolution images. To address these limitations, we propose **CoMemo** - a dual-path architecture that combines a **Co**ntext image path with an image **Memo**ry path for visual processing, effectively alleviating visual information neglect. Additionally, we introduce RoPE-DHR, a novel positional encoding mechanism that employs thumbnail-based positional aggregation to maintain 2D spatial awareness while mitigating remote decay in extended sequences. Evaluations across seven benchmarks,including long-context comprehension, multi-image reasoning, and visual question answering, demonstrate CoMemo's superior performance compared to conventional LVLM architectures. Project page is available at https://lalbj.github.io/projects/CoMemo/.

## 1. Introduction

Recent advances in large language models (LLMs) have demonstrated unprecedented generative capabilities (Ouyang et al., 2022; Touvron et al., 2023), primarily driven by the exponential scaling of training data and model parameters. Building upon this foundation, large vision-language models (LVLMs) have emerged as powerful multimodal systems that align visual representations with LLM embedding spaces to enable cross-modal reasoning (Liu et al., 2024b; Alayrac et al., 2022). Current methodologies for visual information injection predominantly follow two architectural paradigms.

The first paradigm (referred to as LVLM-X, e.g., Flamingo (Alayrac et al., 2022)) employs cross-attention mechanisms to integrate visual features into textual representations. While this approach offers flexible modality interaction, recent studies (Laurençon et al., 2024) have revealed its suboptimal performance compared to alternative approaches when using identical LLM backbones and training data - a finding corroborated by our studies in Figure 1.

The second paradigm (referred to as LVLM-S, e.g., LLaVA (Liu et al., 2024b)) aligns visual tokens into text token embeddings space and then performs autoregressive processing. This paradigm is more compatible with LLM architectures; however, the preservation intrinsic mechanisms such as attention bimodality (Xiao et al., 2023; Liu et al., 2025a) and the linearly increasing position encoding leads to critical limitations: (1) the "lost in the middle" (Liu et al., 2024c; Song et al., 2024) phenomenon degrades performance with increasing context length, and (2) positional encoding sparsity induces remote decay and 2d-dimensional lost in high-resolution image processing.

In this paper, we propose a novel framework for LVLM, named CoMemo. Our key idea is to introduce an additional image-processing mechanism that is unaffected by context, without modifying the internal mechanisms of the LLM. Specifically, we concatenate image tokens with text tokens as the input sequence for fully autoregressive processing, while simultaneously feeding the image tokens into a mixin layer for cross-attention computation. Cross-attention retrieves image information based on text, avoiding the issue of image neglect that can occur in causal self-attention. However, simply combining these two structures, as in NVLM-H (Dai et al., 2024), does not work well. To address this, we first investigated a balanced scheme for visual representation inputs. Then, we introduced a three-stage training technique to prevent overreliance on the cross-attention path, effectively transforming it into a "memory

---
[*]Equal contribution [†]Project lead [1]Shanghai Artificial Intelligence Laboratory [2]Tsinghua University [3]The Chinese University of Hong Kong. Correspondence to: Weijie Su <suweijie@pjlab.org.cn>.

*Proceedings of the 42^{nd} International Conference on Machine Learning*, Vancouver, Canada. PMLR 267, 2025. Copyright 2025 by the author(s).

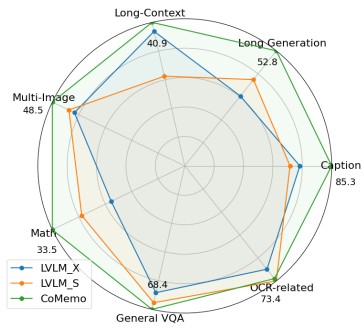

Figure 1: Evaluation results of three architectures with same training data and model size (2B). Please refer to Tables 2 to 4 for details.

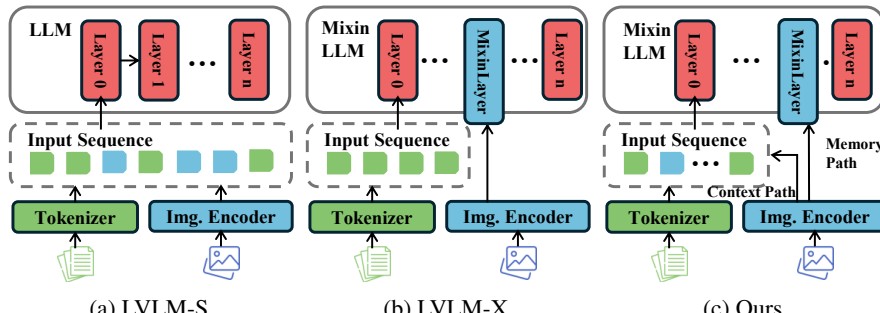

(a) LVLM-S.     (b) LVLM-X.     (c) Ours.

Figure 2: Comparison three types of architectures for LVLMs. Method (a) use image encoder to align visual features with the LLM's continuous token representation space. Method (b) employs mixin layer with cross-attention to update LLM's hidden states based on visual features. And Method (c) contrust a dual-path structure to enable the model to focus more on visual content during generation.

path." Meanwhile, the fully autoregressive process serves as the primary path for introducing image context, referred to as the "context path."

The positional encoding scheme in LVLMs typically adopts RoPE from LLMs, treating each image patch token as an individual token for encoding. However, this approach results in highly sparse positional encodings for dynamic high-resolution image patch tokens. Such sparse encoding can lead to remote decay issues in positional encoding, and the one-dimensional incremental encoding scheme also loses the two-dimensional information of the image. To address this, we propose a novel positional encoding scheme based on the dynamic high-resolution method. Specifically, in the dynamic high-resolution approach, the image is divided into multiple image tiles and a single image thumbnail. We treat the image thumbnail as part of the input sequence for standard sequential positional encoding, while mapping the image tiles to the image thumbnail index based on their two-dimensional positional relationships.

To fully evaluate the impact of the CoMemo architecture, we collected a set of multimodal benchmarks and categorized them into seven evaluation tasks: Caption, Long-generation, Multi-image, Long-context, Math, General VQA, and OCR-related tasks. The results demonstrates CoMemo's superiority over LVLM-X/S baselines under same data and model settings. Our framework achieves 17.2%, 7.0% and 5.6% relative improvement on Caption, Long-Generation and Long-Context tasks respectively, with consistent gains across various benchmarks.

## 2. Design Thinking for CoMemo

### 2.1. Why LVLMs Tend to "lose in the middle"?

Previous studies have shown that both LLMs and LVLMs exhibit the "Lost in the middle" phenomenon (Liu et al.,

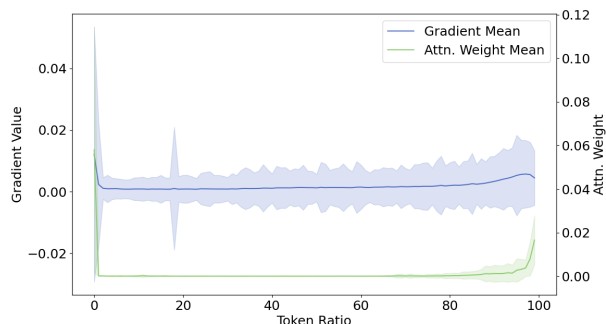

Figure 3: Average gradients and attention weights assigned to tokens at corresponding positions. We computed the average over 1,000 samples.

2024c; Song et al., 2024), as illustrated in Figure 8a. This refers to their struggle to capture key information placed in the middle of the input. However, this phenomenon arises from the causal self-attention and in-context mechanisms inherent in these models. Both LLMs and LVLMs are trained using the next-token prediction paradigm, which heavily relies on contextual tokens during prediction.

As shown in Figure 3, we plot the gradients of each input token during training and the attention weights assigned to each token during inference for the InternVL2 model. To handle sequences of varying lengths, we map each sequence into 100 bins based on the position of each token. A significant portion of the gradient for the current predicted token is backpropagated to nearby tokens. As a result, models trained in this way tend to allocate most of their attention to adjacent tokens during inference, while the initial token, which has a larger gradient and attention, acts as an attention sink to release redundant attention (Xiao et al., 2023). This results in the effective capture of key information at the be-

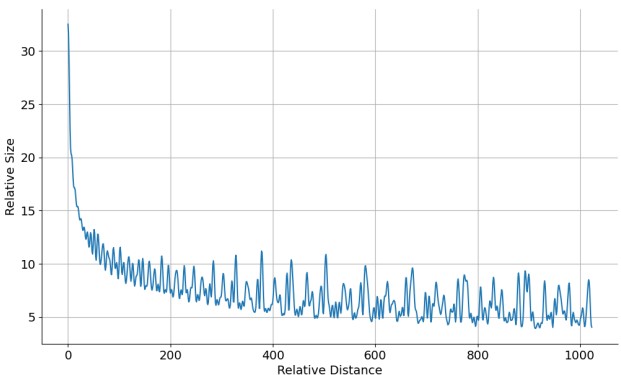

Figure 4: Remote decay estimation for InternVL2-2B. The relative distance refers to the difference between absolute position IDs. In RoPE, the position ID of each input token increments by 1 with the input sequence.

ginning of the sequence, as well as nearby tokens benefiting from contextual attention.

However, key information placed in the middle of the sequence is more likely to be lost. As the context length increases, the middle segment, which is at a higher risk of being lost, also extends.

> **Finding 1:** The attention distribution mechanism in causal self-attention is the cause of the "Lost in the middle" phenomenon. This mechanism also limits the performance of LVLMs in long-context scenarios.

## 2.2. Remote Decay in LVLMs with DHR

Dynamic high resolution significantly enhances the performance of visual tasks by introducing more visual context, especially in tasks that require high resolution, such as OCR-related evaluations. However, this benefit introduces a critical trade-off: excessive visual context exacerbates the remote decay issue in Rotary Position Embedding (RoPE), ultimately limiting model effectiveness in long-context scenarios.

RoPE implements relative position encoding through absolute encoding mechanisms, formally expressed as:

$$(\mathcal{R}_m\boldsymbol{q})^\top(\mathcal{R}_n\boldsymbol{k}) = \mathrm{Re}\left[\sum_{i=0}^{d/2-1}\boldsymbol{q}_{[2i:2i+1]}\boldsymbol{k}^*_{[2i:2i+1]}e^{\mathrm{i}(m-n)\theta_i}\right],$$
(1)

where $m, n$ denote token positions, $d$ the embedding dimension, and $\theta_i$ follows the sinusoidal position encoding scheme. This formulation inherently inherits the remote decay characteristics of sinusoidal encodings.

RoPE exhibits the remote decay property where the relative size between tokens decreases as the relative distance in-

creases (Su, 2021), as shown in Figure 4. While standard InternVL2 processes 256 image tokens per image, activating DHR with a dynamic number of 6 expands this to 1,792 tokens—a 7× increase that further reduces the influence of image tokens during generation.

> **Finding 2:** The context expansion from DHR fundamentally aggravates image neglect.

## 2.3. The Balance Between tow Pathways

CoMemo faces a critical challenge in balancing two visual processing pathways: the cross-attention path and the fully autoregressive path. Through systematic experiments with different high-resolution allocation strategies and training strategies, we identify two key balancing principles. In the mixin layers, we incorporate a gating mechanism that utilizes gate values to reflect the influence of the cross-attention path during the decoding process. Therefore, we choose to evaluate the dependency of LVLM on these two paths from this perspective. Our analysis averages the gates values of mixin layers to quantify pathway preference, with performance evaluated across captioning, general VQA, and OCR tasks.

**Balance in DHR Allocation.** We compare three distinct DHR allocation strategies: (1) allocating DHR information exclusively to the fully autoregressive path, termed DHR-S, (2) assigning it solely to the cross-attention path, termed DHR-X, which corresponds to the allocation mechanism of NVLM-H (Dai et al., 2024), and (3) distributing it to both pathways, termed DHR-B. In unilateral allocation scenarios, the counterpart pathway receives only thumbnail resolution information. As shown in Figure 5, DHR allocation significantly influences pathway specialization. When DHR is exclusively allocated to one pathway, the model shows a pronounced bias toward that pathway. In contrast, dual-pathway allocation results in more stable and balanced

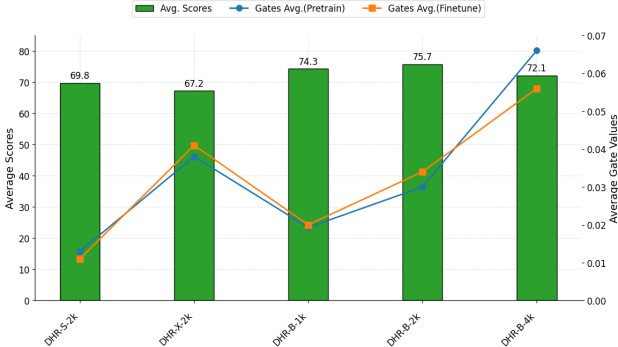

Figure 5: Balancing experiments. Experiment settings are described in Section 2.3. "1k", "2k" and "4k" means pre-train steps. All scores are evaluated after fine-tuning the pretrained checkpoint corresponding to the x-axis.

model behavior.

**Balance in training steps.** Our findings reveal that pretraining steps profoundly affects the equilibrium between the two mechanisms. Although fine-tuning involves extended training (9,000 steps), the visual processing paradigm is largely established during pretraining. In Figure 5, the DHR-B configuration demonstrates that insufficient pretraining steps result in suboptimal alignment, leading to compromised fine-tuning performance. Conversely, excessive pretraining induces over-reliance on specific pathways. While fine-tuning stage attempts to mitigate this imbalance, it ultimately fails to fully rectify the entrenched dependency in "DHR-B-4k" in Figure 5.

This phenomenon arises because during pretraining, only the memory branch and projector parameters are trainable. The projector's limited function of mapping image representations into the text space provides minimal gains in visual comprehension. Consequently, prolonged pretraining naturally reinforces reliance on the cross-attention branch.

> **Finding 3:** CoMemo exhibits a fundamental balancing challenge between dual visual processing pathways.

> **Finding 4:** CoMemo's dual-path visual processing paradigm establishes during pretraining.

## 3. Method

### 3.1. RoPE-DHR

Standard RoPE implementations in LLMs employ a continuous incremental positional encoding scheme, which is inherited by LVLMs. While effective for sequential text data, this approach faces significant limitations when processing high-resolution visual inputs: (1) *remote decay* due to extended context lengths, and (2) *dimensional collapse* of 2D spatial relationships. To address these limitations, we propose RoPE-DHR, a novel position encoding method that employs a hierarchical strategy.

We first processes the thumbnail patch tokens using conventional RoPE to generate base position IDs. For high-resolution tiles, we establish geometric correspondence by mapping each tile patch's coordinates $(x_{tile}, y_{tile})$ to its corresponding thumbnail patch index $i_{thumb}$. This mapping is defined as:

$$i_{thumb} = (\lfloor x_{tile} \times \frac{W_{tile}}{W_{orig}} \rfloor + wb_{tile}, \lfloor y_{tile} \times \frac{H_{orig}}{H_{thumb}} \rfloor + hb_{tile}) \quad (2)$$

where $(W_{orig}, H_{orig})$ and $(W_{tile}, H_{tile})$ denote the original and tile dimensions respectively. The terms $wb_{tile}$ and $hb_{tile}$

represents the start position biases for the tile in the width and height dimensions. This mapping preserves 2D spatial relationships while maintaining compatibility with existing RoPE implementations. In Figure 6, we visualize the raw image, the thumnail and their corresponding patch token position IDs using a color bar.

Crucially, our method decouples positional encoding from absolute sequence positions through: (1) length reduction: prevent the sparse position encoding for DHR by compression the position length in global perspective. (2) geometry reserve: tile patches inherit positional context from their thumbnail anchors.

### 3.2. Architecture of CoMemo

While LLaVA-like architectures demonstrate effective visual-language alignment, they exhibit a tendency to disregard visual information when processing lengthy contextual inputs or generating extended responses. To address this limitation, we propose the CoMemo architecture, which is based on three key structures:

**Dual-stream Structure.** CoMemo maintains tow visual processing streams: (1) *context path* serves as the primary processing stream where image representations are treated as special tokens, concatenated with text tokens to form the input sequence for autoregressive modeling. (2) *memory path* establishes an auxiliary processing stream where image representations interact with the input sequence through cross-attention mechanisms. The two paths maintain identical image representations, ensuring feature consistency while enabling complementary processing and the balance between in tow path as we disccused in Section 2.3.

**Position-aware Cross-attention.** Existing LVLM-X typically employ absolute positional encoding for image patch tokens during encoding (Dubey et al., 2024), with some variants introducing additional positional semantics through specialized tokens (Dai et al., 2024). However, these approaches provide only unidirectional positional awareness. To address this limitation, we implement RoPE in cross-modal attention, establishing bidirectional positional awareness: query positions ($pos_s$) correspond to input sequence token ordering and key positions ($pos_i$) align with visual token indices in the input sequence. The attention mask employs bidirectional visibility constraints, similar to mPlug-owl3 (Ye et al., 2024), where image tokens are visible to their corresponding sequence positions while maintaining bidirectional attention.

**Memory Mixin Strategy** As shown in Figure 7, CoMemo mixin layers are interleaved with standard transformer blocks at a 1:4 ratio. Each memory layer performs: (1) Gated Cross-attention: Modulates visual influence through learnable attention gates ($attn\_gate$). (2) Adaptive Feed-

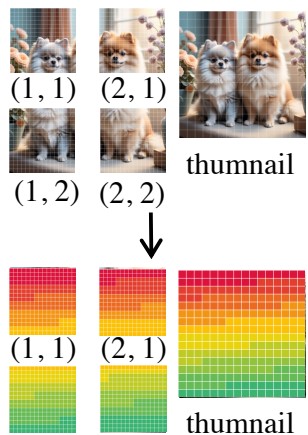

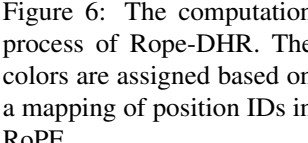

Figure 6: The computation process of Rope-DHR. The colors are assigned based on a mapping of position IDs in RoPE.

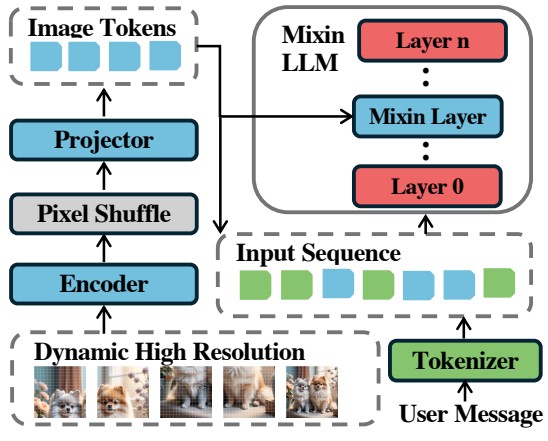

Figure 7: Framework of CoMemo. Both paths share the same encoder and projector. The pixel shuffle technique is adopted from InternVL series (Team, 2024; Chen et al., 2024c).

**Algorithm 1** Mixin Layers

**Require:** $h_s$ (sequence hidden states),
$h_i$ (image hidden states),
$attn\_gate$, $ffw\_gate$,
$pos_s$ (sequence position IDs),
$pos_i$ (image position IDs)

**Ensure:** Updated $h_s$

1: $h_s \leftarrow h_s + \tanh(attn\_gate) \odot$ cross_attn$(q = h_s, kv = h_i, pos_s = pos_s, pos_i = pos_i)$
2: $h_s \leftarrow h_s + \tanh(ffw\_gate) \odot ffw(h_s)$
3: **Return:** $h_s$

forward: Enhances feature transformation via gated nonlinearity (ffw_gate).

During autoregressive decoding, CoMemo requires only a single-step computation between the current decoding token and the cached visual memory states, eliminating the need for key-value caches. This approach circumvents the issue of increasing key-value cache size as the sequence lengthens. The orthogonal design of the architecture ensures compatibility with existing LLaVA variants.

### 3.3. Training Stages

Traditional LVLMs training consists of two phases: pretraining and fine-tuning. During the pretraining phase, we selectively update the projector module and memory architecture parameters while keeping other components frozen. This phase prioritizes cross-modal representation alignment and dynamic equilibrium between dual visual processing pathways.

A critical challenge emerges during pretraining stemming from asymmetric parameter updates: (1) Insufficient training iterations lead to suboptimal projector learning; (2) Prolonged training induces excessive dependency on the memory pathway for LVLM decoding. This imbalance originates from the disparate update strategies - full fine-tuning of memory parameters versus partial tuning of the context projector. Consequently, the LVLM exhibits inherent bias towards memory-based information retrieval to achieve loss minimization.

To mitigate this optimization bias, we propose a three-stage training strategy. In the first stage, we tune the parameters of projector and mixin layers. In the next stage, we freeze the gate parameters. This design aims to enable the model to learn representation alignment and balance the dual-path structure in the first stage. After a certain number of training steps to prevent over-reliance on the cross-attention path, we freeze the corresponding gate control parameters and continue training until the alignment structure is sufficiently learned.

The subsequent fine-tuning stage adopts full-parameter training paradigm. During this stage, all model parameters become trainable with the training objective shifted towards instruction-following.

## 4. Experiments

### 4.1. Setup

To ensure a fair comparison of the capabilities across various architectures, all three of our models utilize the same pretraining and fine-tuning datasets. Each architecture adopts InternLM-1.8B as the LLM backbone and InternViT-300M as the image encoder. The hidden dimensions of each architecture's network, along with hyperparameters such as learning rate and weight decay during training, are also kept consistent. We use the same training data as InternVL-2. Therefore, LVLM-S in our experiments essentially represents InternVL-2. Training for all architectures is divided into two phases: pretraining and fine-tuning. For specific settings, please refer to the appendix.

Table 1: Comparison with other LVLMs of different architectures. * indicates results obtained from our own experiments. Other results are sourced from official reports or third-party evaluation leaderboards.

| Model | Params. | Caption | | | Long-Generation | | Multi-Image | | | Long-Context | | Math | | General VQA | | | OCR-Related | | |
|---|---|---|---|---|---|---|---|---|---|---|---|---|---|---|---|---|---|---|---|
| | | COCO | Flickr | No-Caps | LLaVABen. | MMDU | BLINK | Mantis | MMT | MM-NIAH | MileBench | MathVista | MathVision | MMBench | MME | MMVP | AI2D | ChartQA | TextVQA |
| Closed-Source | | | | | | | | | | | | | | | | | | | |
| GPT-4o | - | - | - | - | - | 70.2 | 68.0 | - | 65.4 | - | 60.3 | 63.8 | - | - | - | - | 84.6 | 85.7 | 77.4 |
| GPT-4V | - | - | - | - | - | - | 54.6 | 62.7 | 64.3 | - | 53.0 | 58.1 | - | - | 1926.6 | 38.7 | 78.2 | 78.5 | 78.0 |
| LVLM-X | | | | | | | | | | | | | | | | | | | |
| mPlug-owl3 | 8B | 90* | 72* | 94.2* | 64.2* | - | 50.3 | 63.1 | - | - | 50.0* | - | - | - | - | - | 73.4 | - | - |
| LLama3.2-V | 11B | - | - | - | 80.9 | - | 39.8 | - | 53.1 | - | - | 51.5 | - | 77.3 | 1820.5 | - | 91.1 | 83.4 | 73.1 |
| LVLM-S | | | | | | | | | | | | | | | | | | | |
| MiniCPM-V-2 | 2.8B | - | - | - | 66.1 | - | 41.2 | - | 54.5 | - | - | 39 | 15.4 | 62.9 | 1808.2 | - | 64.8 | 59.6 | - |
| InternVL-2 | 2B | 79.1* | 65.4* | 60.0* | 62.9* | 28.7* | 38.2* | 48.3* | 50.2* | 27.0* | 52.1* | 48.0* | 16.4* | 73.1* | 1869* | 31.3* | 74.3* | 75.6* | 74.2* |
| InternVL-2.5 | 2B | 114* | 80.2* | 108.6* | - | 33.6* | 44.0 | 54.8 | 54.5 | 22.2* | 50.1* | 51.3 | 13.5 | 74.7 | 2138.2 | 40 | 74.9 | 79.2 | 74.3 |
| Qwen2-VL | 2B | 108.2 | 86.3 | - | 52.5 | - | 44.4 | - | 55.1 | - | - | 43.0 | 19.7 | - | 2326.8 | - | 74.7 | 73.5 | 79.7 |
| CoMemo | 2B | 98.6 | 78.5 | 78.8 | 66.9 | 38.7 | 43.5 | 50.6 | 51.3 | 34.2 | 54.6 | 50 | 17.0 | 74.2 | 1904 | 36.0 | 74.2 | 73.6 | 72.6 |

Table 2: The results on Generation and Math benchmarks. The test settings are described in Section 4.3. The highest scores are highlighted in **bold**.

| Model | Caption | | | Long Generation | | Math | |
|---|---|---|---|---|---|---|---|
| | COCO | Flickr | No-Caps | LLaVABench | MMDU | MathVista | MathVision |
| LVLM-X | 84.9 | 68.9 | 62.9 | 50.8 | 31.5 | 44.2 | 15.8 |
| LVLM-S | 79.1 | 65.4 | 60.0 | 62.9 | 28.7 | 48.0 | 16.4 |
| Ours | **98.6** | **78.5** | **78.8** | **66.9** | **38.7** | **50.0** | **17.0** |

Table 3: The results on Multi-image and Long-context benchmarks. The test settings are described in Section 4.4. The highest scores are highlighted in **bold**.

| Model | Multi-Image | | | Long-Context | | |
|---|---|---|---|---|---|---|
| | BLINK | Mantis | MMT | MM-NIAH-M | MM-NIAH-T | MileBench |
| LVLM-X | 41.5 | 46.5 | 47.8 | **39.5**[1] | 29.5[1] | 53.2[1] |
| LVLM-S | 38.2 | 48.3 | 50.2 | 26.7 | 27.3 | 52.1 |
| Ours | **43.5** | **50.6** | **51.3** | 33.7 | **34.6** | **54.6** |

## 4.2. Comparison with Other LVLMs

As shown in Table 1, we benchmark CoMemo against leading open-source and proprietary LVLMs. It is important to note that our primary objective is architectural ablation rather than absolute performance maximization. The comparison with other models serves to demonstrate that our model was trained on a large-scale dataset and achieves top-tier performance, rather than being tested on toy datasets.

## 4.3. Generation and Math Benchmark

Contemporary multimodal benchmarks primarily rely on constrained response formats (e.g., multiple-choice questions and Yes/No queries) to facilitate evaluation. However, such approaches inadequately assess open-ended reasoning capabilities, as free-form responses introduce higher diversity and complexity. To comprehensively evaluate model capabilities across different granularities, we collected several benchmarks for open-ended evaluation:

- Caption Generation: Evaluates the model's ability to generate concise image descriptions (10-15 words) using CIDEr scores on COCO (Lin et al., 2014a), Flickr30k (Plummer et al., 2015), and No-Caps (Agrawal et al., 2019) datasets.

- Long Generation: Evaluates extended inference using LLaVABench (Liu et al., 2024b) and MMDU (Liu et al., 2024e). LLaVABench focuses on single-image context reasoning, involving a few hundred context tokens, while MMDU is a multi-image analysis with lengthy textual context (avg. 6.4k tokens) and multiple images (ranging from 2 to 20 images).

- Math: Measures model reasoning ability on math diagrams and visual math problems using MathVision (Wang et al., 2024a) and MathVista (Lu et al., 2023). During evaluation, we asked the model to provide step-by-step reasoning and extracted the final answer to calculate accuracy.

As shown in 2, our analysis reveals three key findings. First, LVLM-X's continuous attention mechanism demonstrates superiority in pure visual captioning tasks, achieving a +4% average improvement. Second, LVLM-S's causal attention architecture achieves better performance in knowledge-intensive scenarios, demonstrating enhanced contextual reasoning capabilities from its LLM backbone. Our proposed CoMemo combines the advantages of both approaches, outperforming the original architectures across various tasks. This supports our hypothesis that dual-path attention allocation effectively integrates the benefits of both architectures: maintaining visual grounding while enabling complex reasoning.

## 4.4. Multi-image and Long-context Benchmark

To systematically evaluate multimodal long-context understanding, we establish two complementary evaluation dimensions:

---

[1]LVLM-X's single image token compression reduces average context length by 50% (e.g., 32k→16k).

- Multi-image: When DHR is enabled, each image contributes approximately 2k tokens to the context length, thereby necessitating extended context capacity for multi-image analysis. We evaluate performance on the BLINK (Fu et al., 2025), Mantis (Jiang et al., 2024), and MMT (Ying et al., 2024) datasets.

- Long-context: Tests information extraction in long context scenarios. We select MM-NIAH (Wang et al., 2024c) evaluation that detects image/text needles within hybrid long contexts. And MileBench (Song et al., 2024) progressively challenging tasks with 2-109 images. These benchmarks systematically quantify long-context capabilities from both textual token and visual token perspectives.

As shown in Table 3, our proposed architecture achieves the best performance in scenarios involving multiple images and long contexts. Specifically, MM-Niah-T represents the needle that is the key information placed in the text, while MM-NIAH-M represents the needle placed in the image. In the evaluation of MM-NIAH-T, the memory structure stores image data that is unrelated to the needle, redundant information. Nevertheless, our model still achieves the best performance. This not only demonstrates that compressing the image token position space through RoPE-DHR enhances the model's ability to understand long sequence texts but also indicates that the Memory structure does not cause the model to overfocus on image information, thereby preserving its ability to retrieve and reason with language effectively.

In the MileBench evaluation, due to the potential for excessive image tokens leading to long context sequences and out-of-memory issues, we did not enable DHR settings. Therefore, in this evaluation, each image in the input sequence has only a single image thumbnail. In this scenario, our architecture's positional encoding is the same as LVLM-S, primarily reflecting the role of the memory structure. Despite this, our architecture still achieved a 2.5% improvement over LVLM-S.

The MileBench benchmark also includes needle in haystack tasks for both text and image scenarios. In Figure 8, we visualize the average results for these two types of NIAH tasks in MileBench. As mentioned earlier, we observed that the attention mechanism of LLMs and LVLMs exhibits a bimodal distribution, where LVLMs tend to focus more on the beginning and the most recent tokens, leading to the "Lost in the middle" phenomenon. This means that when the needle is placed in the middle of the sequence, the model's performance on NIAH tasks deteriorates. Our architecture, however, addresses this issue by continuously focusing on the "middle image information" during the token generation process, effectively mitigating this problem.

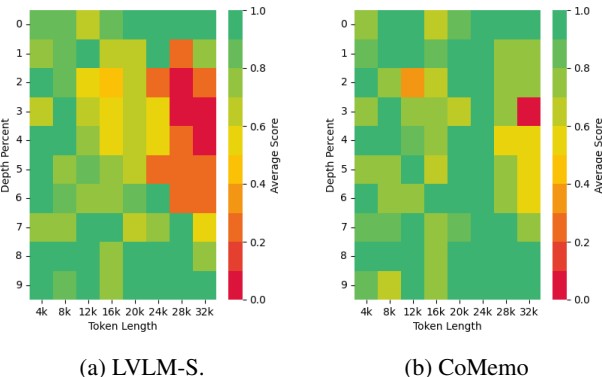

|                | (a) LVLM-S. | (b) CoMemo |

Figure 8: Heatmap of results for the NIAH evaluation on MileBench benchmark. The depth percentage indicates the position of the target information (needle) relative to the entire sequence.

Table 4: The results on General VQA and OCR-related benchmarks. The test settings are described in Section 4.5. The highest scores are highlighted in **bold**.

| Model | General VQA | | | OCR-Related | | |
|-------|---------|------|------|------|---------|--------|
|       | MMBench | MME  | MMVP | AI2D | ChartQA | TextVQA |
| LVLM-X | 69.6 | 1812 | 27.3 | 69.9 | 69.4 | 68.8 |
| LVLM-S | 73.1 | 1869 | 31.3 | **74.3** | **75.6** | **74.2** |
| Ours | **74.2** | **1904** | **36.0** | 74.2 | 73.6 | 72.6 |

## 4.5. General VQA and OCR Benchmark

To comprehensively evaluate CoMemo's multimodal understanding capabilities, we conduct extensive experiments on conventional vision-language benchmarks covering two main categories:

- General VQA: Assessing visual perception and reasoning abilities through single-image question answering, including MMBench (Liu et al., 2025b), MME (Fu et al., 2023), and MMVP (Zhong et al., 2023).

- OCR-Related: Requiring fine-grained information extraction from diagrams and charts, evaluated on AI2D (Hiippala et al., 2021), TextVQA (Singh et al., 2019b), and ChartQA (Masry et al., 2022b).

The input sequences in these two types of benchmarks are relatively short, and responses typically consisting of single words or short phrases. In contrast, CoMemo incorporates a memory structure that alleviates image neglect by introducing an additional image focus mechanism. Additionally, RoPE-DHR addresses image neglect by compressing image information to reduce the long-range decay caused by positional encoding. While these techniques are not specifically tailored for the benchmarks mentioned above, our architecture still performs competitively when compared to

LVLM-S.

As shown in Table 4, our architecture performs slightly better than LVLM-S on some general multimodal benchmarks. However, in tasks such as text and OCR, which require high-resolution image information, traditional approaches typically rely on more granular image representations. This approach contrasts with the philosophy underlying our approach, which focuses on improving the model's long-context and long-generation capabilities.

### 4.6. Training and Inference Efficiency Comparison

Table 5: Training efficiency comparison across different 2B parameter models.

| Model | Batch size | # of A100 GPUs | Train steps/s | Train samples/s |
|---|---|---|---|---|
| LVLM-X | 1024 | 64 | 0.123 | 15.71 |
| LVLM-S | 1024 | 64 | 0.105 | 13.4 |
| CoMemo | 1024 | 64 | 0.096 | 12.26 |

Table 6: Inference speed comparison across different 2B parameter models (lower is better).

| Model | Batch size | # of A100 GPUs | Caption COCO (sec) | MMBench (sec) | MMNIAH (min) | TextVQA (sec) |
|---|---|---|---|---|---|---|
| LVLM-X | 1 | 8 | 260 | 76 | 22 | 88 |
| LVLM-S | 1 | 8 | 270 | 90 | 25 | 100 |
| CoMemo | 1 | 8 | 280 | 110 | 30 | 120 |

To comprehensively compare the training and inference efficiency of three architectures, we measured their sample throughput during training and inference times across multiple benchmarks. As shown in Table 5 and Table 6, we report the training and inference efficiency for each architecture. The results demonstrate that CoMemo achieves latency comparable to LVLM-S. Although LVLM-X exhibits higher efficiency due to the use of fewer image tokens, its performance is significantly inferior to both CoMemo and LVLM-S.

### 4.7. Ablation Study

**Ablation on Components of CoMemo.** We conducted a complete ablation study on components using a 2B-scale model, as shown in Variants 1 to 5 in Table 7. When only RoPE-DHR is introduced, the compressed position encoding significantly improves performance on both long-generation and long-context tasks (Variant 2). When only the memory path was introduced, it addressed issues related to neglecting image information, leading to some improvement in the model's performance (Variant 3). However, since the image tokens in the memory path lack positional information, during cross-attention computation, there is no positional correspondence between image tokens and text tokens. Moreover, the lack of distinguishing features between image tiles, thumnails, and multiple images hindered the model's capabilities in different dimensions. Therefore,

after incorporating RoPE-DHR, the model's capabilities in different dimensions were further enhanced (Variant 5). However, since RoPE-DHR is essentially a compression-oriented encoding, it may affect scenarios like OCR that require fine-grained information.

**Ablation on compression ratio of RoPE-DHR.** In the main experiments, RoPE-DHR uses shared position IDs for image tokens in the thumbnail and their corresponding subimage tokens. This approach effectively compresses the position encoding. Therefore, we propose a variant, RoPE-DHR without compression, where the position encodings for subimage tokens corresponding to a two-dimensional position increment by 1 relative to the thumbnail image tokens, while the position IDs between thumbnail image tokens increment based on the number of tokens at their corresponding positions, rather than by 1. The experimental results are shown in Table 7 in Variants 4. It can be observed that, without compression, the model outperforms the LVLM-S architecture across all dimensions.

**Ablation on different scale models.** To verify that CoMemo adheres to the scaling law, we select InternLM-7B as the language model for experiments at the 7B scale. As shown in Table 7 in Variant 6 and 7, at the 8B scale, CoMemo's average performance across all dimensions remains superior to the LVLM-S architecture. The CoMemo architecture continues to deliver outstanding performance in both Cation and Long-context tasks. As language models scale up, the impact of compressed position encoding becomes more pronounced in OCR tasks.

**Consistency on Different Datasets.** We also conducted dataset-switching experiments using the open-source InternVL-1.2 1.2M fine-tuning dataset (OpenGVLab, 2024) during the SFT stage, while keeping the pretraining data consistent. As shown in Variants 8 to 9 in Table 7, even with the changes in the dataset, our CoMemo consistently outperforms the LVLM-S architecture across various dimensions.

## 5. Related Works

### 5.1. Mainstream LVLMs and Their Architectures

Contemporary LVLMs typically employ pre-trained language models as decoders, utilizing two dominant strategies for visual-text alignment: (1) cross-attention mechanisms and (2) joint projector-autoregressive architectures.

Fully Autoregressive LVLMs: LLaVA (Liu et al., 2024b) pioneered this approach by projecting image representations into the LLM's space and jointly decoding them with text tokens. Subsequent models like VILA-v1.5 (Lin et al., 2024) and LLaVA-Next (Liu et al., 2024a) built on this architecture, with LLaVA-Next introducing dynamic high-resolution techniques for improved performance. Other ad-

Table 7: Ablation Study. The test settings are described in Section 4.7. "NC" means to use RoPE-DHR without compression.

| Name | Params. | Memory | RoPE-DHR | Datasets | Overall | Caption | Long-Generation | Multi-Image | Long-Context | Math | General VQA | OCR-Related |
|---|---|---|---|---|---|---|---|---|---|---|---|---|
| Variant 1 | 2B | ✗ | ✗ | Ours | 55.4 | 68.1 | 45.8 | 45.5 | 39.5 | 32.2 | 65.9 | 74.7 |
| Variant 2 | 2B | ✗ | ✓ | Ours | 56.7 | 67.2 | 51.4 | 47.5 | 42.0 | 31.7 | 69.2 | 72.9 |
| Variant 3 | 2B | ✓ | ✗ | Ours | 59.0 | 82.8 | 49.6 | 46.0 | 43.0 | 32.2 | 66.7 | 75.6 |
| Variant 4 | 2B | ✓ | NC | Ours | 59.5 | 79.7 | 51.9 | 48.5 | 43.1 | 34.7 | 67.3 | 74.8 |
| Variant 5 | 2B | ✓ | ✓ | Ours | 60.5 | 85.3 | 52.8 | 48.5 | 44.4 | 33.5 | 68.4 | 73.4 |
| Variant 6 | 8B | ✗ | ✗ | Ours | 65.4 | 77.7 | 61.0 | 57.4 | 45.3 | 38.7 | 78.3 | 82.3 |
| Variant 7 | 8B | ✓ | ✓ | Ours | 68.0 | 92.1 | 61.4 | 57.7 | 50.8 | 38.9 | 78.1 | 79.1 |
| Variant 8 | 2B | ✗ | ✗ | 1.2M | 51.1 | 79.7 | 36.1 | 44.1 | 28.7 | 28.8 | 58.5 | 61.8 |
| Variant 9 | 2B | ✓ | ✓ | 1.2M | 54.6 | 89.5 | 38.4 | 46.2 | 31.5 | 28.7 | 62.3 | 63.5 |

vancements include Qwen2-VL (Wang et al., 2024b), which introduced M-RoPE, and InternVL-2.5 (Chen et al., 2024c), which used pixel shuffle to reduce token count after DHR. DeepSeek-VL2 (Wu et al., 2024) employed a pretrained MoE-based LLM. However, this alignment approach inherits the LLM's generation mechanism, which can lead to issues such as "image neglect" or the "lost in the middle" problem.

Cross-Attention-Based LVLMs: Flamingo (Alayrac et al., 2022) is an early example of LVLMs using cross-attention mechanisms. Later models, like Idefics (Laurençon et al., 2024) and LLaMa-3.2-Vision (Dubey et al., 2024), adopted its mixin layer design, which introduces cross-attention and gating mechanisms. EVLM (Chen et al., 2024b) experimented with using intermediate Vision Transformer representations as inputs to the mixin layer. mPlug-owl3 (Ye et al., 2024) added adaptive gating and hyper-layer fusion to combine cross-attention and self-attention. This approach enables visual understanding while maintaining the LLM's frozen language ability, as seen in LLaMa-3.2-Vision. However, in these models, image representations are aligned directly to the LLM's hidden state, whereas LLaVA-like methods align them with the text token space, better leveraging autoregressive decoding capabilities and improving performance.

### 5.2. Position Encoding Schemes in LVLMs

Most LVLMs use position encoding methods inherited from LLMs, primarily RoPE. In these models, each image patch token is treated like a text token and assigned position IDs for RoPE computation. However, several advancements address specific challenges. MiniCPM-V (Yao et al., 2024) introduced an absolute position encoding for each image tile in the context of DHR, while LLaMA-3.2-V (Yao et al., 2024) designed encodings for both image tiles and patch tokens. NVLM (Yao et al., 2024), also leveraging DHR, added special tokens before each tile to convey positional information. While effective for predefined DHR ratios, these methods lack scalability.

In contrast, Qwen-VL2 (Wang et al., 2024b) introduced M-RoPE, a multi-dimensional position encoding extending RoPE to three channels (temporal, height, width) for images and videos. However, this position encoding requires

a customized ViT and thus cannot be applied to LVLMs employing DHR.

Our proposed RoPE-DHR, based on 1D principles, offers a 2D-aware encoding scheme that addresses these challenges without the extra computational burden.

## 6. Conclusion

We present CoMemo, a novel architecture for Large Vision-Language Models specifically designed for long-form generation and extended context understanding. Our approach features a dual-path image processing mechanism and introduces RoPE-DHR to alleviate remote decay in DHR scenarios while restoring critical two-dimensional spatial information. These innovations significantly enhance model performance across multiple tasks including image captioning, long-form generation, long-context understanding, multi-image analysis, and general visual question answering. We hope this work will contribute to the advancement of the vision-language modeling community.

## Impact Statement

This paper presents work whose goal is to advance the field of Large Vision-Language Model. There are many potential societal consequences of our work, none which we feel must be specifically highlighted here.

## Acknowledgments

The work is supported by the National Key R&D Program of China (NO. 2022ZD0161301), by the National Natural Science Foundation of China (U24A20325, 62321005, 62376134).

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

# A. Theoretical Derivation of Remote Decay

**Definition A.1** (RoPE-induced Inner Product). Given a query vector $q \in \mathbb{C}^d$ and a key vector $k \in \mathbb{C}^d$, their inner product under Rotary Position Embedding (RoPE) is defined as:

$$(R_m q)^\top (R_n k) \triangleq \mathrm{Re}\left[ \sum_{i=0}^{d/2-1} q_{[2i:2i+1]} k^*_{[2i:2i+1]} e^{\mathrm{i}(m-n)\theta_i} \right]$$

where $q_{[2i:2i+1]}$ denotes the $i$-th 2D subvector of $q$, $\theta_i$ is the preset rotational frequency, and $^*$ represents complex conjugation.

Building on Definition A.1, we establish the following core lemma:

**Lemma A.2** (Abel Transform Representation). *Let $h_i \triangleq q_{[2i:2i+1]} k^*_{[2i:2i+1]}$ and $S_j \triangleq \sum_{i=0}^{j-1} e^{\mathrm{i}(m-n)\theta_i}$, with boundary conditions $h_{d/2} = 0$ and $S_0 = 0$. Then:*

$$\sum_{i=0}^{d/2-1} h_i e^{\mathrm{i}(m-n)\theta_i} = \sum_{i=0}^{d/2-1} h_i(S_{i+1} - S_i) = - \sum_{i=0}^{d/2-1} S_{i+1}(h_{i+1} - h_i)$$

*Proof.* By Abel summation by parts:

$$\sum_{i=0}^{d/2-1} h_i(S_{i+1} - S_i) = \sum_{i=0}^{d/2-1} h_i S_{i+1} - \sum_{i=0}^{d/2-1} h_i S_i$$

$$= \sum_{i=1}^{d/2} h_{i-1} S_i - \sum_{i=0}^{d/2-1} h_i S_i \quad \text{(index shift)}$$

$$= - \sum_{i=0}^{d/2-1} S_i(h_i - h_{i-1}) + h_{d/2-1} S_{d/2} - h_{-1} S_0$$

$$= - \sum_{i=0}^{d/2-1} S_{i+1}(h_{i+1} - h_i) \quad \text{(using boundary conditions)}$$

$\square$

**Theorem A.3** (Decay Rate Bound). *Under the assumptions of Lemma A.2, the absolute value of the inner product satisfies:*

$$\left| \sum_{i=0}^{d/2-1} h_i e^{\mathrm{i}(m-n)\theta_i} \right| \leq \left( \max_{0 \leq i \leq d/2-1} |h_{i+1} - h_i| \right) \sum_{i=0}^{d/2-1} |S_{i+1}|$$

*Proof.* From Lemma A.2 and the triangle inequality:

$$\left| \sum_{i=0}^{d/2-1} h_i e^{\mathrm{i}(m-n)\theta_i} \right| = \left| \sum_{i=0}^{d/2-1} S_{i+1}(h_{i+1} - h_i) \right|$$

$$\leq \sum_{i=0}^{d/2-1} |S_{i+1}| \cdot |h_{i+1} - h_i|$$

$$\leq \left( \max_i |h_{i+1} - h_i| \right) \sum_{i=0}^{d/2-1} |S_{i+1}|$$

$\square$

## B. Training Details

The hyperparameters used for pretraining and finetuning across the three architectures are listed in Table 5. We observed that the 2B model, with its smaller hidden dimension, requires less extensive training, making Phase 2 optional. Therefore, in the ablation experiments in the main text, we only used Pretraining Phase 1 for the 2B model, while the 8B model utilized both pretraining phases.

Table 5: **Hyperparameters for Training and Inference**

| Parameter | Pretraining Phase 1 | Pretraining Phase 2 | Finetuning |
|---|---|---|---|
| Max sequence length | 8192 | 8192 | 8192 |
| Max tile/image | 12 | 12 | 12 |
| Optimizer | AdamW | AdamW | AdamW |
| Learning rate | $1 \times 10^{-4}$ | $1 \times 10^{-4}$ | $4 \times 10^{-5}$ |
| Weight decay | 0.01 | 0.01 | 0.01 |
| Optimizer momentum | $\beta_1, \beta_2 = 0.9, 0.999$ | $\beta_1, \beta_2 = 0.9, 0.999$ | $\beta_1, \beta_2 = 0.9, 0.999$ |
| Learning rate schedule | Constant with warmup | Constant with warmup | Cosine decay |
| Warmup ratio | 0.03 | 0.03 | 0.03 |
| Training steps | 2000 | 2000 | 9000 |
| Batch size | 1024 | 1024 | 1024 |
| Number of mixin layers | 4 | 4 | 4 |
| Trainable weights | LVLM-S: MLP | | All |
| | LVLM-X: Mixin layers + MLP | | |
| | CoMemo: Mixin layers + MLP | | |
| | (Freeze gate in Phase 2) | | |

## C. Detailed Expereiment Results

We provide the detailed ablation study results in Table 9.

Table 9: Detailed results on albation study.

| Model | Caption | | | Long-Generation | | Multi-Image | | | Long-Context | | Math | | General VQA | | | OCR-Related | | |
|---|---|---|---|---|---|---|---|---|---|---|---|---|---|---|---|---|---|---|
| | COCO | Flickr | No-Caps | LLaVABen. | MMDU | BLINK | Mantis | MMT | MM-NIAH | MileBench | MathVista | MathVision | MMBench | MME | MMVP | AI2D | ChartQA | TextVQA |
| Variant 1 | 79.1 | 65.4 | 60.0 | 62.9 | 28.7 | 38.2 | 48.3 | 50.2 | 27.0 | 52.1 | 48 | 16.5 | 73.1 | 1869 | 31.3 | 74.3 | 75.6 | 74.2 |
| Variant 2 | 74.1 | 63.5 | 64.0 | 64.0 | 38.8 | 42.7 | 48.8 | 51.1 | 30.2 | 53.8 | 48.1 | 15.2 | 72.8 | 1899 | 40 | 73.4 | 72.5 | 72.9 |
| Variant 3 | 98.0 | 74.4 | 75.7 | 65.2 | 34.1 | 37 | 51.6 | 49.4 | 31.9 | 54.2 | 50.1 | 14.3 | 73.1 | 1834 | 35.3 | 75.9 | 76.2 | 74.6 |
| Variant 4 | 90.2 | 67.2 | 81.9 | 64.2 | 39.6 | 42.7 | 52.1 | 50.6 | 30.6 | 55.6 | 49.9 | 19.4 | 74.0 | 1826 | 36.7 | 74.2 | 76.1 | 74.2 |
| Variant 5 | 98.6 | 78.5 | 78.8 | 66.9 | 38.7 | 43.5 | 50.6 | 51.3 | 34.2 | 54.6 | 50.0 | 17.0 | 74.2 | 1904 | 36 | 74.2 | 73.6 | 72.6 |
| Variant 6 | 90.1 | 73.0 | 69.9 | 73.7 | 48.2 | 49.2 | 65.0 | 57.9 | 29.8 | 60.8 | 58.3 | 19.1 | 79.1 | 2210 | 45.3 | 83.5 | 84.6 | 78.7 |
| Variant 7 | 102.0 | 82.2 | 92.2 | 76.2 | 46.5 | 49.7 | 65.4 | 58.0 | 38.2 | 63.4 | 59.7 | 18.1 | 79.8 | 2182 | 45.3 | 83.5 | 77.2 | 76.6 |
| Variant 8 | 92.0 | 61.7 | 85.6 | 42 | 30.2 | 40.5 | 44.2 | 47.8 | 15.6 | 41.9 | 39.9 | 17.7 | 63.8 | 1674 | 28 | 63.2 | 63.8 | 58.6 |
| Variant 9 | 100.5 | 71.5 | 96.6 | 44.8 | 32.0 | 40.6 | 48.8 | 49.2 | 17.5 | 45.6 | 44.2 | 13.2 | 67.6 | 1759 | 31.3 | 63.3 | 64.7 | 62.6 |

## D. Dataset Details

The data used in the pre-training stage are listed in Table 6. And datasets used for instruction tuning are listed in Table 7.

Table 6: Summary of datasets used in the pretraining stage.

| task | dataset |
|---|---|
| Short Caption | Laion (en&zh) (Schuhmann et al., 2022a), COYO (Byeon et al., 2022), COCO (Lin et al., 2014b) |
| OCR | Wukong-OCR (Gu et al., 2022), LaionCOCO-OCR (Schuhmann et al., 2022b) |
| Detection | GRIT (Peng et al., 2023), Objects365 (Shao et al., 2019) |
| Conversation | All-Seeing (en&zh) (Wang et al., 2023b) |
| Image-text instruction data | (see Table 7) |

Table 7: Summary of datasets used in the instruction tuning stage.

| task | dataset |
|---|---|
| General QA | VQAv2 (Goyal et al., 2017), GQA (Hudson & Manning, 2019), OKVQA (Marino et al., 2019), VSR (Liu et al., 2023a) |
| Science | AI2D (Kembhavi et al., 2016), ScienceQA (Lu et al., 2022a), Chemistry Data (Li et al., 2024) |
| | TQA (Kembhavi et al., 2017) |
| Medical | PMC-VQA (Zhang et al., 2023a), VQA-RAD (Lau et al., 2018), VQA-Med (Ben Abacha et al., 2019) |
| | Medical-Diff-VQA (Hu et al., 2023), PathVQA (He et al., 2020), |
| | SLAKE (Liu et al., 2021), PMC-CaseReport (Wu, 2023) |
| Chart | ChartQA (Masry et al., 2022a), LRV-Instruction (Liu et al., 2023b), PlotQA (Methani et al., 2020) |
| | Unichart (Masry et al., 2023), MMC-Inst (Liu et al., 2023c), DVQA (Kafle et al., 2018) |
| | TableMWP (Lu et al., 2022b), FigureQA (Kahou et al., 2017), MapQA (Chang et al., 2022) |
| | SciTSR (Chi et al., 2019), Fintabnet (Zheng et al., 2021) |
| Mathematics | CLEVR (Johnson et al., 2017), MetaMath (Yu et al., 2023), GeoQA+ (Cao & Xiao, 2022) |
| | Geometry3k (Lu et al., 2021), GeoS (Seo et al., 2015), Unigeo (Chen et al., 2022) |
| | Super-CLEVR (Li et al., 2023), MathQA (Amini et al., 2019) |
| Knowledge | Art500k (Mao et al., 2017), MovieNet (Huang et al., 2020), KonIQ-10k (Hosu et al., 2020) |
| | KVQA (Shah et al., 2019), ViQuAE (Lerner et al., 2022) |
| OCR | InfoVQA (Mathew et al., 2022), TextVQA (Singh et al., 2019a), ArT (Chng et al., 2019) |
| | CASIA (Liu et al., 2011), Chart-to-text (Kantharaj et al., 2022), COCO-text (Veit et al., 2016) |
| | CTW (Yuan et al., 2019), EATEN (Guo et al., 2019), ICDAR2019-LSVT (Sun et al., 2019) |
| | ICPR MTWI (He et al., 2018), NAF (Davis et al., 2019), ReCTS (Zhang et al., 2019) |
| | TextOCR (Singh et al., 2021), LLaVAR (Zhang et al., 2023b), HME-100k (Yuan et al., 2022) |
| | POIE (Kuang et al., 2023), SROIE (Huang et al., 2019), ST-VQA (Biten et al., 2019) |
| | EST-VQA (Wang et al., 2020), IAM (Marti & Bunke, 2002) |
| Document | DocVQA (Clark & Gardner, 2017), DocReason25k (Hu et al., 2024) |
| Grounding | RefCOCO (Kazemzadeh et al., 2014), RefCOCO+ (Kazemzadeh et al., 2014), RefCOCOg (Kazemzadeh et al., 2014) |
| | RD-BoxCoT (Chen et al., 2023) |
| Conversation | ALLaVA (Chen et al., 2024a), LAION-GPT4V (LAION, 2023) |
| | MMDU (Liu et al., 2024d), TextOCR-GPT4V (Carter, 2024) |
| Detection | Objects365 (Shao et al., 2019), V3Det (Wang et al., 2023a) |

