# OpenReview forum: "CoMemo: LVLMs Need Image Context with Image Memory"
_ICML.cc/2025/Conference — ICML 2025 poster_

### Official Review · Reviewer_t4ai · 2025-03-10

**Overall Recommendation:** 2

**Summary:**

- This paper proposes CoMemo, a hybrid architecture of LLaVA and Flamingo’s cross-attention module.
- It also has a 2D positional encoding mechanism (RoPE-2D) to better preserve spatial relationships, addressing the issue of visual information neglect in long contexts.
- Experimental results across multiple benchmarks indicate that CoMemo improves performance on tasks requiring long-context comprehension and multi-image reasoning.

**Claims And Evidence:**

The authors claim their contributions are equally split (50% each), but both are weak:

1. Novelty of CoMemo: The authors present CoMemo as a novel architecture, but it is essentially a hybrid of LLaVA and cross-attention (Flamingo-style). While hybridity itself isn't an issue, the same architecture was introduced earlier, e.g., Q-Former in BLIP-2 [1] (published January 2023), which is neither cited nor included in comparisons.
2. Extending RoPE to 2D: The authors claim to be the first to extend RoPE to 2D, but this is incorrect. The concept was already introduced as M-RoPE in Qwen2-VL (published September 2024), which they cite and include in Table 1. They attempt to differentiate their work in lines 412–418, arguing:

    > "In contrast, Qwen-VL2 (Wang et al., 2024b) introduced M-RoPE, a multi-dimensional position encoding extending RoPE to three channels (temporal, height, width) for im- ages and videos. However, this increases computational costs, tripling the original RoPE overhead and introducing redundancy by requiring these channels for text tokens."
    >

    However, this claim is incorrect. The Qwen-VL2 paper explicitly states that when processing 2D images, the temporal dimension remains constant, meaning no additional computational cost:

    > "When processing images, the temporal IDs of each visual token remain constant, while distinct IDs are assigned to the height and width components based on the token’s position in the image."
    >
3. Additionally, the evaluation is weak, with few baselines, missing values in Table 1, and unclear figures (e.g., missing axis ranges in Figure 1). Overall, the work requires significant improvement.

**Essential References Not Discussed:**

The following key literatures are missing:

[1] Li, J., Li, D., Savarese, S. and Hoi, S., 2023, July. Blip-2: Bootstrapping language-image pre-training with frozen image encoders and large language models. In International conference on machine learning (pp. 19730-19742). PMLR.

[2] Li, J., Li, D., Xiong, C., and Hoi, S. Blip: Bootstrapping language-image pre-training for unified vision-language understanding and generation. In International conference on machine learning, pp. 12888–12900. PMLR, 2022.

**Experimental Designs Or Analyses:**

- Table 1 has too many missing values. For example, Qwen2-VL-2B explicitly reports an MMMU score of 41.1 in its paper, but the authors omit this unfavorable result.
- The comparison includes an outdated version of MiniCPM (V-2); the appropriate comparison for InternVL-2.5 should be MiniCPM V2.5 or V2.6.
- Too few baselines are considered. For instance, BLIP [2] and BLIP-2 [1], which are closely related, should be included.

**Methods And Evaluation Criteria:**

Lines 51–52 state: "masks out part of the image information in the mixin layer to prevent over-reliance on this path," raising the following questions:

- How is the masked information selected?
- Since the approach first introduces image context and then applies random dropout, doesn’t this introduce inefficiency? Was this design choice specifically considered?
- Why mask the cross-attention path (MixinLayer) while treating the autoregressive path as the primary method for incorporating image context, rather than the reverse?

**Other Comments Or Suggestions:**

- Figure 1 is difficult to interpret without axis scales.
- If the Mixin layer is essentially cross-attention, renaming it adds unnecessary complexity.
- **Typo:** l.137 – *"tow pathways"* → *"two pathways"*
- **Missing reference:** l.155 – *"As shown in 5"* should specify the correct figure number.

**Other Strengths And Weaknesses:**

Despite the weaknesses in theoretical claims, literature coverage, and evaluation results, the overall writing is poor and unpolished, resulting in broken logic and a lack of clarity.

**Questions For Authors:**

See above.

**Relation To Broader Scientific Literature:**

See below the ‘Essential References Noe Discussed’ section

**Theoretical Claims:**

- It is unclear how Findings 1–3 in Section 2 are derived.
- The main claims are questionable; see the *Claims and Evidence* section above for details.

---

> ### Author Rebuttal · Authors · 2025-04-01
>
> `Q1: CoMemo is essentially a hybrid of LLaVA and cross-attention (Flamingo-style). `
>
> First, neither open-source nor closed-source models currently offer such an architecture. As Reviewer vTuL noted, "It is exciting to see an effective solution that leverages the advantages of both approaches." Furthermore, in L45-46 of our paper, we emphasize that a naive combination of these two structures would fail. It is precisely the novel techniques we introduce—beyond simple hybridization—that differentiate CoMemo from merely hybrid cross-attention and LLaVA framework.
>
> `Q2: The same architecture was introduced earlier, e.g., Q-Former in BLIP-2.`
>
> The most fundamental reason is that the Q-Former is essentially a projector. In BLIP, the representations between text tokens and image tokens are still updated via self-attention, which differs significantly from CoMemo’s approach of directly using cross-attention for representation updates. Therefore, BLIP is not orthogonal to LVLM-S—in essence, it is a special case of the LVLM-S architecture.
>
> `Q3: The authors claim to be the first to extend RoPE to 2D, but this is incorrect. M-RoPE means no additional computational cost.`
>
> This comment misinterprets our claim and overlooks an important technical distinction. As stated in L26–L32 and L82–L84 of our paper, we introduce a RoPE-based 2D encoding scheme specifically designed for the DHR (Dynamic High-Resolution) technique.
>
> Key differences from M-RoPE are as follows:
>
> 1. LLM Training Requirements: M-RoPE operates in 3D space, requiring dedicated training. In contrast, our RoPE-2D uses 1D computation, directly reusing the original RoPE from the LLM without additional alignment training.
>
> 2. Resolution Schemes: M-RoPE was designed for Naive Dynamic Resolution, not DHR. In Qwen-VL, RoPE replaces ViT’s absolute position encoding to support varying resolutions, but this requires extra training.
>
> 3. Additional RoPE Cache Computation: M-RoPE’s 3D RoPE requires cache computation for each dimension, which adds complexity.
>
> `Q4: Missing values in Table 1.`
> This table is only intended to show that our experimental data is not from a small-scale dataset. Additionally, we have now completed the [most missing values](https://drive.google.com/file/d/1jCKPKPMcAjnJc_1FCwqQsSN9W9luK7qP/view?usp=sharing).
>
> `Q5: Qwen2-VL-2B explicitly reports an MMMU score of 41.1 in its paper, but the authors omit this unfavorable result.`
> The benchmark we evaluated does not include MMMU.
>
> `Q6: The appropriate comparison for InternVL-2.5 should be MiniCPM V2.5 or V2.6.`
> Since the CoMemo model in our main experiments is at the 2B scale, and MiniCPM V2.5 and V2.6 are 7B-scale LVLMs, we chose to compare with MiniCPM-V-2, which is closer in model size.
>
> `Q7: Too few baselines are considered. For instance, BLIP [2] and BLIP-2 [1], which are closely related, should be included.`
>
> We selected the two baselines because they are currently the only two mainstream architectures in this field. As for why we did not choose BLIP as a baseline, there are two main reasons:
>
> 1. The BLIP architecture can be considered a refined form of the LLaVA-like architecture and should essentially be covered under the LVLM-S framework.
>
> 2. Previous studies have shown that, with the same data scale, the performance of Q-Former in BLIP is inferior to that of MLP[1, 2]. In BLIP-3, Q-Former is no longer used[3].
> [1] DeCo: Decoupling Token Compression from Semantic Abstraction in Multimodal Large Language Models
> [2] Honeybee: Locality-enhanced Projector for Multimodal LLM
> [3] xGen-MM (BLIP-3): A Family of Open Large Multimodal Models
>
> `Q8: Questions about mask during pretrain.`
> `Q8.1: How is the masked information selected?`
>
> We use a random masking strategy (L261–267) to remove image information by masking image tokens post-tokenization.
>
> `Q8.2: Since the approach first introduces image context and then applies random dropout, doesn’t this introduce inefficiency?`
>
> L186–189 clarify that the Memory Path and Context Path reuse the same image representations, incurring no extra computation. The minor latency from dropout (masking) in the Memory Path is negligible.
>
> `Q8.3: Why mask the cross-attention path while treating the autoregressive path as the primary method?`
>
> In L13–20, we briefly introduced and cited findings from Idefics-2, indicating that under the same parameter count and training data, the LVLM-S approach outperforms LVLM-X.
> In Section 2.3 of the paper, we further conducted an ablation study on this. As shown in Figure 5, as the number of pretraining steps increases, the model exhibits stronger reliance on the cross-attention path, yet this leads to suboptimal performance.

---

### Official Review · Reviewer_PsDG · 2025-03-12

**Overall Recommendation:** 3

**Summary:**

The paper introduces CoMemo to improve visual information retention in multimodal tasks. Specifically, CoMemo includes a memory path for image tokens that operates independently of the main text path. This helps prevent visual information loss during long-context reasoning. Then CoMemo uses RoPE-2D encoding that maintains 2D spatial relationships in high-resolution images. This reduces performance degradation in long sequences. Finally, CoMemo uses the memory mixin strategy during training to ensure both the context path and memory path contribute effectively.

**Claims And Evidence:**

This paper is well-written, systematically introducing the limitations of existing models before presenting effective solutions. For instance, to illustrate the "lost in the middle" problem, the authors provide clear evidence through gradient heatmaps and evaluation results on long-context benchmarks, demonstrating CoMemo's improved performance over LVLM-X and LVLM-S.

**Essential References Not Discussed:**

The references are comprehensive and well-organized.

**Experimental Designs Or Analyses:**

The experimental setup is reasonable and the authors give sufficient analysis.

**Methods And Evaluation Criteria:**

The evaluation process used in this paper follows a popular setup.

**Other Comments Or Suggestions:**

“2.3. The balance between tow pathways” -》 “2.3. The balance between two pathways”

**Other Strengths And Weaknesses:**

* RoPE-2D’s compression may degrade performance in fine-grained tasks like OCR.
* The dual-path architecture and RoPE-2D introduce additional computational overhead, potentially impacting real-time applications.

**Questions For Authors:**

* RoPE-2D’s compression may degrade performance in fine-grained tasks like OCR. This doesn't seem very reasonable. Does the authors have any opinion on it?
* The authors show that LVLMs have the "lost in the middle" phenomenon with gradient and attention weights in Fig 3. I would like to know if the proposed method truly improves these aspects and if it can provide clearer visual results.

**Relation To Broader Scientific Literature:**

The authors introduce their proposed method by addressing three key challenges: the "lost in the middle" phenomenon, "remote decay in dynamic high-resolution models", and "the balance between two pathways." The "lost in the middle" issue has been identified in previous studies [1,2], while "remote decay in dynamic high-resolution models" is a well-recognized challenge in the field.

[1] Liu, N. F., Lin, K., Hewitt, J., Paranjape, A., Bevilacqua,M., Petroni, F., and Liang, P. Lost in the middle: How language models use long contexts.
[2] Song, D., Chen, S., Chen, G. H., Yu, F., Wan, X., and Wang, B. Milebench: Benchmarking mllms in long context.

**Theoretical Claims:**

The theoretical claims presented in the paper are solid.

---

> ### Author Rebuttal · Authors · 2025-04-01
>
> `Q1: The dual-path architecture and RoPE-2D introduce additional computational overhead, potentially impacting real-time applications.`
>
> While CoMemo introduces some computational overhead, we have designed mechanisms to minimize latency, making the time difference between CoMemo and LVLM-S nearly negligible.
>
> 1. Shared image token representation: Both the memory and context paths share the same image token representations, so the ViT image encoding is performed once without adding latency.
>
> 2. Fewer cross-attention layers: Unlike Flamingo and MLLama-3.2, which insert cross-attention after each transformer block, we use a 4:1 ratio, inserting only 6 cross-attention layers in InternLM2 (24 layers in total). Cross-attention computes interactions only between input tokens and image tokens, reducing computation since the image token sequence is shorter than the input sequence.
>
> 3. No KV-cache required for cross-attention: Cross-attention eliminates the need for key-value caching during inference. Decoding only requires computing the query for the current token, whereas self-attention involves additional memory for KV-cache and quadratic time complexity (O(N²)).
>
> As shown in [Table 4](https://drive.google.com/file/d/1hcnKbAzDiZnCSX9GFJY2l2jObVkDYcDa/view?usp=sharing) and [Table 5](https://drive.google.com/file/d/1MPYN2gf7sCnyZA8os0DGaSsDidYKqxX2/view?usp=sharing), we report training and inference efficiency across different architectures. The results confirm that CoMemo has nearly the same latency as LVLM-S. Although LVLM-X achieves higher efficiency due to using fewer image tokens, its performance is significantly weaker than both CoMemo and LVLM-S.
>
> `Q2: RoPE-2D’s compression may degrade performance in fine-grained tasks like OCR.`
>
> This is an important point, and we did not provide a detailed discussion in the original manuscript. The compression in RoPE-2D is a trade-off between improving long-context and long-generation tasks while potentially degrading performance on fine-grained tasks like OCR.
> We propose a variant, RoPE-2D (no-overlap), which removes the compression. This version shows consistent improvements across various benchmarks, including OCR tasks.
> Why does RoPE-2D affect OCR performance?
>
> 1. OCR is a fine-grained visual task that requires detailed information. The compression of positional information in RoPE-2D can reduce performance on tasks needing high resolution.
>
> 2. In OCR evaluations, excessive compression may occur, especially when the dynamic number for DHR (set based on InternVL and VLMEvalKit) is large. For example, in ChartQA, the compressed positional encoding causes 3328 image tokens to correspond to only 256 position IDs.
>
> What if we map positional relationships without compression?
>
> We explored RoPE-2D (no-overlap), where:
>
> 1. Subimage position IDs are mapped to thumbnail patch positions.
>
> 2. Position IDs accumulate based on the number of mapped patch tokens.
> In this variant, position IDs are incrementally assigned, ensuring uniqueness across subimages. As shown in [Table 2](https://drive.google.com/file/d/1lXmFL1Nl0YYsIu6FStLt7v7VjMFUm1f5/view?usp=sharing), this approach improves all evaluation metrics compared to the baseline. However, it performs worse on long-context and generation tasks compared to the compressed RoPE-2D version.
> We will include a more detailed analysis of RoPE-2D in the next version.
>
> `Q3: The authors show that LVLMs have the "lost in the middle" phenomenon with gradient and attention weights in Fig 3. I would like to know if the proposed method truly improves these aspects and if it can provide clearer visual results.`
>
> Thank you for this insightful question. We analyze the “lost in the middle” phenomenon from two perspectives and evaluate CoMemo’s improvements in these areas:
>
> 1. Attention Weights: CoMemo retains the self-attention mechanism from LVLM-S, so the image attention remains largely unchanged. However, CoMemo introduces an additional cross-attention mechanism, allowing input tokens to attend to image tokens, providing explicit visual grounding. As discussed in Section 2.3, the average gate value quantifies the strength of this visual attention, which is unaffected by the input context and thus does not suffer from the “lost in the middle” issue.
>
> 2. Image Token Gradients: In [Figure 1](https://drive.google.com/file/d/1gMQOcUC4qkBur4RVMLw6PbDCUC8JtQt0/view?usp=sharing), we compare the average gradients of image tokens during inference between LVLM-S and CoMemo across different benchmarks. We compute the gradients of output logits with respect to input image tokens, take the absolute value, and average across all image tokens. The results show that CoMemo significantly strengthens visual grounding. On the MMNIAH benchmark, where the “lost in the middle” issue is prominent, image token gradients in CoMemo nearly double compared to LVLM-S, clearly indicating that CoMemo mitigates this problem.
>
> We will correct the typos issue in the next version.

---

### Official Review · Reviewer_HPwo · 2025-03-14

**Overall Recommendation:** 4

**Summary:**

This paper thoroughly investigates the flaws of LLM architectures when processing multimodal inputs, including the progressive neglect of central visual content as context expands and the failure of conventional positional encoding schemes in preserving 2D structures. To address these issues, this paper presents CoMemo to decouple the memory path for mitigating visual neglect and RoPE-2D to maintain 2D spatial awareness. The experimental results indicate the effectiveness of the proposed methods in multiple benchmarks.

###update after the rebuttal#####

Thanks to the authors for responding to my arguments in this paper. The newly added experimental results in the provided link are sufficient. It addressed my concerns. I tend to raise my score.

############################

**Claims And Evidence:**

The claims are reasonable.

**Essential References Not Discussed:**

NA

**Experimental Designs Or Analyses:**

The authors only employ a small-scale model, i.e., InternLM-1.8B, to conduct experiments, lacking the verification of the universality of the proposed schemes. The authors are advised to verify the effectiveness across multiple architectures to ensure reliability.

**Methods And Evaluation Criteria:**

The employed benchmarks are comprehensive and reasonable for evaluating performance.

**Other Comments Or Suggestions:**

Missing a space between words in the abstract part.

Please pay attention to keeping consistent decimal places in experimental results.

**Other Strengths And Weaknesses:**

This paper is well-organized, and the listed key findings are insightful.

**Questions For Authors:**

1. In Table 1, the reviewer noticed that lots of results are missing for the majority of the models. What is the reason?
2. The performance of CoMemo still has significant gaps compared with other models on certain benchmarks. What is the reason?
3. Can the authors provide more explicit analyses and visualizations to indicate the effective mechanism of the proposed methods?

**Relation To Broader Scientific Literature:**

NA

**Theoretical Claims:**

NA

---

> ### Author Rebuttal · Authors · 2025-04-01
>
> Thank you very much for your recognition of our work and your positive comments regarding the well-orgnized and insightful findings of our paper.
> Below, we will address your concerns point by point, and all suggested revisions will be incorporated into the next version of the manuscript. If our responses adequately address your concerns, we would sincerely appreciate it if you could consider adjusting your evaluation score. Thank you again for your time and thoughtful review.
>
> `Q1: The performance of CoMemo still has significant gaps compared with other models on certain benchmarks. What is the reason?`
>
> The main reason for CoMemo's performance gap compared to other models is the training data. As seen in the evolution of open-source models like InternVL, QwenVL, and LLaVA, architectural improvements were largely driven by data strategies, rather than changes in architecture. Differences in training data can obscure architectural distinctions, complicating the selection of the optimal model. As discussed in Section 4.2, the goal of this paper is to conduct an ablation study on model architectures using the same dataset, not to maximize absolute performance. Comparing with SOTA models highlights that our training dataset is not based on toy datasets, demonstrating the generalization ability of our results.
>
> `Q2: In Table 1, the reviewer noticed that lots of results are missing for the majority of the models. What is the reason?`
>
> We apologize for the confusion caused by the missing values in Table 1. Due to time constraints, we were unable to conduct a comprehensive evaluation, but we ensured that most benchmarks have at least three or more models for comparison in this [url](https://drive.google.com/file/d/1jCKPKPMcAjnJc_1FCwqQsSN9W9luK7qP/view?usp=sharing). Additionally, we will consider including a comparison of different models at the 8B scale in the next version.
>
> `Q3: The authors only employ a small-scale model, i.e., InternLM-1.8B, to conduct experiments, lacking the verification of the universality of the proposed schemes.`
>
> Thank you for your constructive and insightful suggestions. We agree that evaluating our framework at larger scales can better demonstrate its effectiveness. Therefore, we have conducted additional experiments at the 7B scale see [Table 3](https://drive.google.com/file/d/1O6jIbebOjUmgPlfTAa1ytDVRuY_1IDTt/view?usp=sharing).
>
> For the 7B-scale model, we adopted InternLM2-7B as the language backbone and InternViT-300M as the visual encoder. The results at this scale are largely consistent with those observed in the 2B experiments, further validating that our architecture follows the scaling law. As shown, performance improved across most vision tasks at the 7B scale. However, we observed a slight drop in performance on OCR-related tasks, which we attribute to the compression characteristics of our RoPE-2D positional encoding.
>
> To address this issue, we also propose a variant called RoPE-2D (no-overlap). Our experiments at the 2B scale show that this variant provides more stable improvements across different tasks in [Table 2](https://drive.google.com/file/d/1lXmFL1Nl0YYsIu6FStLt7v7VjMFUm1f5/view?usp=sharing). Due to time and resource constraints—training a 7B-scale model on our dataset requires nearly two days with 128 A100 GPUs—we were unable to explore more positional encoding strategies or conduct further large-scale experiments at this time. However, we plan to include more extensive evaluations to further demonstrate the generalizability of our method in the next version.
>
> `Q4: Can the authors provide more explicit analyses and visualizations to indicate the effective mechanism of the proposed methods?`
>
> In [Figure 1](https://drive.google.com/file/d/1gMQOcUC4qkBur4RVMLw6PbDCUC8JtQt0/view?usp=sharing), we compare the average gradients of input image tokens between LVLM-S and CoMemo during inference across different benchmarks. Specifically, we compute the gradients with respect to the input tokens based on the logits corresponding to the model's response token IDs, then extract the image tokens from the indexed results. Since we only consider the magnitude of influence, we take the absolute value of the computed gradients before averaging across all image tokens to obtain the final result.
>
> The comparison shows that through architectural adjustments and positional encoding modifications, the model's responses indeed demonstrate stronger focus on visual information, which verifies our mitigation of the image neglect problem.
>
> `Q5: Missing a space between words in the abstract part. Please pay attention to keeping consistent decimal places in experimental results.`
>
> Thank you for pointing out the formatting issues in our paper. We will address these problems and make the necessary corrections in the next version.

---

### Official Review · Reviewer_vTuL · 2025-03-17

**Overall Recommendation:** 3

**Summary:**

This paper introduces CoMemo, provides two key design choices into MLLMs, 1) adding additional cross-attn like Flamingo, besides the origianl llava-style approach, but to mask part of the info to prevent over-reliance 2) add 2d rope to LLM backbone for image features. The approach ourperforms llava-style and flamingo-style. The author also provides detailed study into MLLMs: 1) study of the lost in the middle phenomenon 2) The remote decay of rope 3) the balance between two pathways

**Claims And Evidence:**

See below

**Essential References Not Discussed:**

N/A

**Experimental Designs Or Analyses:**

See below

**Methods And Evaluation Criteria:**

See below

**Other Comments Or Suggestions:**

There is a long standing debate between the flamingo-style and llava-style, It's very exciting to see a good solution to leverage both advantages. The experiments results are good, while the paper is not ready, and the writing should be improved. I think with major revision, this paper can be a very good and impactful work, but the current submission is not there yet.

**Other Strengths And Weaknesses:**

The writing is very unclear, and it's a bad experience to read this paper, an example is that the authors mentioned attn_gate and ffw_gate at Section 2.3, but first introduce it in 3.2. The writing of Balance in DHR Allocation is also unclear, the author should point out that 1k, 2k, 4k means steps. and I still dont know what is the detailed calculation of gates avg. Algo1 should be format using a smaller font size, now the line-break is very weird.
The experiments are solid, and the results are promising, the prosoed methods consistently outperforms -X and -S variants, this paper provides a promising solution to integrate flamingo and llava style MLLMs.

**Questions For Authors:**

To test free-form responses, the author can also test more populat benchmark like mm-vet, can authors provide justifications on this point.
It seems the rope-2d is orthogonal to the -S variant, can CoMemo still outperforms -S variant in Table 2, Table3 and Table4 if applying rope-2d to -S variant.
It seems the rope-2d is harmful to OCR in Table5, also in Table 4, CoMemo is worse than -S variant, Could you provide some justification on this point?

**Relation To Broader Scientific Literature:**

N/A

**Theoretical Claims:**

See below

---

> ### Author Rebuttal · Authors · 2025-04-01
>
> Thank you for your valuable feedback to improve our work's clarity. As you rightly pointed out, proposing a framework that combines Flamingo-style and LLaVA-style approaches represents a highly impactful contribution. Thank you for recognizing the value of our work.
> Below, we address your concerns point by point, and all suggested revisions will be incorporated in the next version. If you feel we have adequately addressed your feedback, we would be most grateful if you could consider adjusting your evaluation score accordingly. Thank you for your time and consideration.
>
> `Q1: I still dont know what is the detailed calculation of gates avg.`
>
> In L143 of the original manuscript, we have already mentioned how the avg. gates are calculated:
> > Our analysis averages the attn_gate and ffw_gate values to quantify pathway preference.
>
> Specifically, the average gates value is obtained by calculating the mean of the absolute values of the gate values from each mixin layer, corresponding to the data shown in the line chart in Figure 5.
>
> `Q2: Authors mentioned attn_gate and ffw_gate at Section 2.3, but first introduce it in 3.2.`
> This is because attn_gate and ffw_gate are not novel concepts introduced in our paper. However, we believe that adding citations when first mentioning these terms, or restructuring the content could significantly reduce potential confusion.
>
> `Q3: The writing of Balance in DHR Allocation is also unclear, the author should point out that 1k, 2k, 4k means steps.`
> They appear in Figure 5, as we stated in the caption of Figure 5:
> > pretrained checkpoint corresponding to the x-axis.
>
> The values 1k, 2k, and 4k indeed denote to the training steps during pretraining.
>
> `Q4: Algo1's format`
>
> We apologize for the poor reading experience caused by the formatting issue. In the next version, we will adjust the layout to alleviate the impact of the line-breaks on readability.
>
> `Q5: The author can also test more populat benchmark like mm-vet`
>
> Sure! As evident from the [Table 4](https://drive.google.com/file/d/1_OsHi8-Ahy1YhaYpS6NK98q7ZOE-N0QM/view?usp=sharing), since MMVet is a small-scale evaluation set (200 Q&A pairs), CoMemo does not significantly outperform LVLM-S in overall score. However, it still demonstrates superiority in certain dimensions emphasizing visual perception (Recognition/Spatial/Math).
>
> `Q6: It seems the rope-2d is harmful to OCR in Table5, Could you provide some justification on this point?`
>
> We have previously addressed this trade-off in the manuscript (L361-367, L392-394). However, we now propose an improved RoPE-2D(no-overlaps) variant that achieves more balanced improvements across all capabilities.
>
> Why RoPE-2D affects OCR performance? The performance trade-off occurs because we compress positional encoding IDs to mitigate the remote decay problem. While this compression significantly benefits tasks that don't require fine-grained image information, it inevitably reduces information richness for OCR tasks. For instance, in ChartQA evaluations with a maximum number of 12, our compression maps 3,328 image tokens to only 256 RoPE position IDs.
>
> The computation of RoPE-2D (no-overlaps) involves the following two steps:
>
> 1. Subimage position IDs are mapped to their corresponding thumbnail patch positions.
> (In the initial version of RoPE-2D, all subimage position IDs matched their corresponding thumbnail patch IDs. Here, however, they are sequentially incremented starting from the thumbnail patch IDs.)
>
> 2. Thumbnail position IDs accumulate based on the number of mapped patch tokens
> As shown in [Table 2](https://drive.google.com/file/d/1lXmFL1Nl0YYsIu6FStLt7v7VjMFUm1f5/view?usp=sharing), this approach demonstrates improvements across all evaluation metrics compared to the baseline (LVLM-S vs CoMemo+RoPE2D(no-overlaps)). However, it shows significantly weaker performance on long-context and generation tasks compared to the compressed RoPE-2D version.
>
> `Q7: Can CoMemo still outperforms -S variant if applying rope-2d to -S variant.`
>
> Yes, RoPE-2D can indeed be applied to the -S variant.
> As shown in Table 2, our 2D positional encoding reconstruction enhances LVLM-S's performance across multiple capabilities: multi-image processing, general VQA, and image captioning tasks. This approach also improves long-context understanding and extended text generation by mitigating the remote decay issue through compressed positional encoding. However, as previously noted, we observed some degradation in OCR performance. Due to time constraints, we were unable to conduct experiments with the RoPE-2D(no-overlaps) variant. Based on the consistent improvements demonstrated in Table 2, we anticipate this variant would similarly provide stable performance gains across various capabilities.
>
> We appreciate the constructive comments from the reviewer regarding RoPE-2D. We believe that these experiments and analyses will help followers gain a more comprehensive understanding of RoPE-2D.

---

> > ### Comment · Reviewer_vTuL · 2025-04-03
> >
> > Thank you for authors response, I apologize for time reason I do not cover all details in the paper earlier. My concern has been resolved and I will update my score to 3. Hope authors can polish the writing before camera ready for better reading experience.

---

> > > ### Author Response · Authors · 2025-04-07
> > >
> > > Thanks for raising the scores! We have polished the areas that were unclear as mentioned in the reviewer comments. We will also add the additional experiments from the rebuttal stage to the next version and continue refining the text, which will help clarify the motivation and strengths of our method.

---

### Decision · Program_Chairs · 2025-05-01

**Decision:**

Accept (poster)

**Comment:**

This paper introduces CoMemo, a dual-path architecture addressing the "lost in the middle" issue in VLM. This extra branch processes image tokens through a mixin layer to preserve central visual details. It also introduces a RoPE‑2D to enhance 2D spatial relationships. Experiments show this approach enhances long-context reasoning and multi-image understanding in LVLMs.

The paper received 2 weak accept, 1 accept, and 1 weak reject. After carefully read the paper, the reviews and the rebuttal/feedback, the ACs decided to put the paper as `weak accept`, considering the following points can be improved further:

1. Presentation clarify, as pointed out by reviewer `vTuL`.

2. The training data details seem not provided; not sure if the comparison with the other 2B models is fair.

3. The selection of benchmarks appears limited and selective, which weakens the convincingness of the reported improvements; expanding the evaluation to include a wider range of tests—and incorporating more 7B-scale model results—would better demonstrate the effectiveness and scalability of the proposed modules.